# 6DoF Pose Estimation of Transparent Object from a Single RGB-D Image

**DOI:** 10.3390/s20236790

**Published:** 2020-11-27

**Authors:** Chi Xu, Jiale Chen, Mengyang Yao, Jun Zhou, Lijun Zhang, Yi Liu

**Affiliations:** 1School of Automation, China University of Geosciences, Wuhan 430074, China; xuchi@cug.edu.cn (C.X.); vtasyao@163.com (M.Y.); jchow@cug.edu.cn (J.Z.); lijunzh@cug.edu.cn (L.Z.); 2Hubei Key Laboratory of Advanced Control and Intelligent Automation for Complex Systems, Wuhan 430074, China; 3Engineering Research Center of Intelligent Technology for Geo-Exploration, Ministry of Education, Wuhan 430074, China; 4CRRC Zhuzhou Electric Locomotive Co., Ltd. 1 TianXin Road; Zhuzhou 412000, China; liuyi_hust@163.com; 5National Innovation Center of Advanced Rail Transit Equipment, Zhuzhou 412000, China

**Keywords:** 6Dof pose estimation, transparent object, human-computer interaction

## Abstract

6DoF object pose estimation is a foundation for many important applications, such as robotic grasping, automatic driving, and so on. However, it is very challenging to estimate 6DoF pose of transparent object which is commonly seen in our daily life, because the optical characteristics of transparent material lead to significant depth error which results in false estimation. To solve this problem, a two-stage approach is proposed to estimate 6DoF pose of transparent object from a single RGB-D image. In the first stage, the influence of the depth error is eliminated by transparent segmentation, surface normal recovering, and RANSAC plane estimation. In the second stage, an extended point-cloud representation is presented to accurately and efficiently estimate object pose. As far as we know, it is the first deep learning based approach which focuses on 6DoF pose estimation of transparent objects from a single RGB-D image. Experimental results show that the proposed approach can effectively estimate 6DoF pose of transparent object, and it out-performs the state-of-the-art baselines by a large margin.

## 1. Introduction

6DoF (Degrees of Freedom) pose estimation aims at estimating an object’s rotation (3DoF) and translation (3DoF) in the camera coordinate frame [1,2,3]. In some papers, “6DoF” is also referred to as “6D” for short. It is a key technology closely related to many important real-world applications, such as robotic grasping [4,5], automatic driving [6,7], augmented reality [8,9], and so on. With the emergence of consumer-level RGB-D sensor (e.g., Kinect, Intel RealSense, etc.), 6DoF object pose estimation accuracy has been significantly boosted by using RGB-D image [2,3,10,11].

Transparent objects (e.g., glasses, plastic bottles, bowls, etc.) are commonly seen in our daily environments, such as kitchen, office, living room, canteen, and so on. However, existing pose estimation methods for ordinary objects cannot deal with transparent ones correctly (please refer to Figure 1), because the optical characteristics of transparent material lead to significant depth error [12] in D-channel of RGB-D image (the D-channel encodes the depth from the object to the camera, which contains important hint to retrieve the 3D geometric information of the observed object). As can be seen in Figure 2, the depth error of transparent object can be classified into two types: (i) missing depth, i.e. the depth of specific regions is missing due to specular reflection on surface of transparent object; and (ii) background depth, i.e., instead of the true depth on object surface, the false distorted depth on background behind the object is captured, since light passes through the transparent material and refraction occurs. The cause of the depth error is illustrated in Figure 3. The above-mentioned depth errors distort the 3D geometric information of the transparent object observed, and it significantly degrades the pose estimation accuracy of transparent object.

In this paper, we propose an accurate and efficient approach for 6DoF pose estimation of transparent object from a single RGB-D image. As illustrated in Figure 4, the proposed approach contains two stages: In the first stage, we eliminate the influence of depth errors by transparent segmentation, surface normal recovering, and RANSAC plane estimation. In the second stage, an extended point cloud representation is constructed based on the output of the first stage, and the color feature is extracted from cropped color patch. The extended point-cloud and the color feature extracted are then fed into a DenseFusion-like network structure [2] for 6DoF pose estimation. The block-diagram of our approach is shown in Figure 5. The RGB-channel is fed into the transparent segmentation module to retrieve bounding box and mask of transparent object. Then, the color feature is extracted, the surface normal is recovered, and the plane is estimated. By taking the normal, the plane, and the UV map as inputs, the extended point-cloud is constructed. Finally, based on the color feature and the extended point-cloud, the 6DoF object pose is estimated.

The recovered surface normal, the plane, and the UV map are essential components of the extended point-cloud representation, which contains rich geometric information for 6DoF pose estimation: (1) the recovered surface normal contains important hint to estimate the object’s relative pose (3DoF rotation); (2) the plane where the object placed is closely related to the object’s 3D position; and (3) the UV map encodes the 2D coordinates of points on image, which is crucial for 6DoF object pose estimation [1,13].

The work most related to our approach is ClearGrasp [12]. Similar to Sajjan et al. [12], we estimate surface normal of transparent object. Different from Sajjan et al. [12], we focus on 6DoF pose estimation, while ClearGrasp aims at depth reconstruction.

Our approach is very different from the direct feeding of the reconstructed depth into an ordinary RGB-D pose estimator, as in [12]. The differences are mainly in the following to two aspects: (1) ClearGrasp [12] reconstructs depth using a global optimization scheme which is time consuming, but we do not reconstruct depth so that the inferring efficiency is largely accelerated. (2) The ordinary RGB-D pose estimator takes the depth as input, and the depth is converted into classic point-cloud for pose estimation. However, our approach does not rely on classic point-cloud, therefore depth reconstruction is not required.

Overall, the contributions of this paper are twofold:We propose a new deep learning based approach which focuses on 6DoF pose estimation of transparent object from a single RGB-D image. Discriminative high-level deep features are retrieved through a two-stage end-to-end neural network, which result in an accurate 6DoF pose estimation.We introduce a novel extended point cloud representation for 6DoF pose estimation. Different from classic point cloud, the representation does not require depth as input. With this representation, object pose can be efficiently recovered without the time consuming depth reconstruction.

Experimental results show that the proposed approach significantly outperforms state-of-the-art baselines in terms of accuracy and efficiency.

## 2. Related Work

This paper focuses on 6DoF pose estimation of transparent object from a single RGB-D image. In this section, we review the related literature regarding the following aspects:

**Traditional object pose estimation**. Traditional methods primarily utilize hand-crafted low-level features for 6DoF object pose estimation. In [14,15], oriented point pair is used to describe the global information of object model. In [16], a template matching method is proposed to detect 3D objects. In [17], a RANSAC-based scheme is designed to randomly match three correspondences between the scene and the object model. As the traditional methods rely on low-level features, they are not as accurate as deep learning based methods.

**Deep learning based object pose estimation**. In recent years, deep learning has largely improved the accuracy and robustness of 6DoF object pose estimation, as high-level deep features are much more discriminative than low-level traditional features. Early works [10,11] directly apply 2D CNN on RGB-D image for 6DoF pose regression, while 2D CNN does not characterize 3D geometric information well. To better explore the 3D geometric information, the 3D space corresponding to depth image is divided into voxel grids, and then 3D CNN is applied on voxels [18,19,20]. Higher-dimensional convolution on 3D voxel requires huge computational resources. To improve the computational efficiency, point-net based methods [21,22] directly extract deep geometric features from classic point-cloud while retaining the computational efficiency. DenseFusion [2] further augments the point-cloud based geometric features by embedding the corresponding color features, and it significantly improves the pose estimation accuracy. Based on the geometric and color features of Wang et al. [2], Rotation Anchor [3] uses a discrete-continuous formulation for rotation regression to resolve local-optimum problem of symmetric objects. Nevertheless, the above-mentioned ordinary pose estimators cannot correctly deal with transparent object.

**Detection of transparent objects**. Transparent object is a very difficult topic in computer vision field. The appearance of transparent objects can vary dramatically because of the reflective and refractive nature of transparent material. Many research works focus on transparent object detection. Fritz et al. [23] proposed an additive latent feature for transparent object recognition. McHenry et al. [24] identified glass edges by training hierarchical SVM. Philips et al. [25] segregated semi-transparent objects by a stereoscopic cue. Xie et al. [26] proposed a boundary-aware approach to segment transparent object. In [27], glass regions are segmented using geodesic active contour based optimization. In [28], glass objects are localized by joint inferring boundary and depth. In [29,30,31], transparent objects are detected using CNN networks. However, these methods only detect transparent object, and do not estimate 6DoF object pose.

**3D reconstruction of transparent objects**. Depth can be reconstructed from constrained background environments. In [32,33], transparent objects are reconstructed with known background patterns. In [34,35,36], the depth of transparent objects is reconstructed using time of flight camera, since glass absorbs light of certain wavelengths. 3D geometry can be reconstructed from multiple views or 3D scanning. Ji et al. [37] conducted a volumetric reconstruction of transparent objects by fusing depth and silhouette from multiple images with known poses. Li et al. [38] presented a physically based network to recover 3D shape of transparent object from multiple color images. Albrecht et al. [39] reconstructed the geometry of transparent object from point cloud of multiple views. Different from the above works, Sajjan et al. [12] reconstructed the depth as well as the surface normal of transparent object from a single RGB-D image. However, the 6DoF pose estimation is still not addressed in [12].

**Pose estimation of transparent objects**. Estimating pose of transparent object from a single RGB-D image is a challenging task. Some works estimate pose using traditional features. In [40,41], the pose of a rigid transparent object is estimated by 2D edge feature analysis. In [42], SIFT features are used to recognize the transparent object. However, low-level traditional features are not as discriminative as high-level deep features. To the best of our knowledge, in previous work, high-level deep features have not been utilized to estimate 6DoF transparent object pose from a single RGB-D image. It is worth noting that other sensors can also be used for transparent object pose estimation. For example, transparent object pose can be estimated through a monocular color camera [43,44], but the translation estimation along the z-axis tends to be inaccurate due to lack of 3D depth information. Stereo camera [45,46], light field camera [47], single pixel camera [48], and microscope–camera system [49] can be used for object pose estimation, but these works are very different from this paper and are not discussed further.

## 3. Method

This work aims to predict the 6DoF pose of known transparent object in the camera coordinate frame using a single RGB-D image. The object pose is represented by a 3D rigid transformation (R,T) with rotation R∈SO(3) and translation $T\in R^{3}$. Normally, there exist multiple object instances within one image. For each segmented object instance, the proposed approach estimates the object pose through a two-stage framework, as shown in Figure 4. In the first stage, there are three modules: transparent segmentation, surface normal estimation, and RANSAC plane estimation. The results of the first stage are fed into the second stage for further processing. In the second stage, there are three modules: extended point-cloud, color feature extraction, and 6DoF pose estimation. These six modules are the key subsections of our approach. Details are described as follows.

### 3.1. The First Stage

The first stage takes a single RGB-D image as input. The RGB-D image contains two parts: RGB-channel and D-channel. The RGB-channel is used to segment the 2D transparent region and recover the 3D surface normal. The D-channel is used to estimate the 3D plane where the object is placed. The roles of the modules are as follow: the transparent segmentation module identifies the transparent region and segments the object instances; the surface normal estimation module and the RANSAC plane estimation module together recover the transparent object’s 3D geometric information.

**Transparent segmentation.** The transparent object instance is identified by the transparent segmentation module. We segment the region of interest corresponding to transparent object instance through a Mask R-CNN [50] network. Given a single RGB-D image, its RGB-channel is fed into the segmentation network as input, and the output is a list of detected object instances and the corresponding segmentation maps. The transparent region is segmented for three reasons: (1) to remove the noisy depth of transparent region and keep the reliable depth of non-transparent region; (2) to calculate 2D bounding box and crop image patches for further feature extraction; and (3) to sample extended point-cloud on transparent object.

Mask R-CNN [50] is an accurate and mature component of many 6DoF pose estimation methods, such as DenseFusion [2] and Rotation Anchor [3]. In this paper, the Mask R-CNN component is trained and evaluated using the standard scheme, as in previous works [2,3,50]. The number of classes is 5 in our experiments. Mask R-CNN is very accurate for transparent segmentation. To investigate how much the performance of Mask R-CNN impacts the rest of the pipeline, we evaluate the whole pipeline using the Mask R-CNN segmentation and the ground-truth segmentation, respectively. The results show that the 6DoF pose estimation accuracy based on the ground-truth segmentation is only 0.7% higher than that based on the Mask R-CNN segmentation. Besides, other instance segmentation networks (such as SegNet [51] and RefineNet [52]) can also be adopted for transparent segmentation.

**Surface normal recovering**. Our surface normal estimation network adopts an encoding–decoding structure which is the same as that of Sajjan et al. [12]. The network structure is shown in Figure 6. Firstly, taking the RGB-channel as input, one convolution layer and several residual blocks are used to obtain low-level features. Secondly, the Atrous Spatial Pyramid Pooling [53] sub-network is used to sample high-level features. It not only captures the context in multi-scale but also retains the global characteristics of feature through average pooling. We apply skip connections between the low-level features and the high-level features to ensure the integrity. Beside, we use many residual blocks with dilated convolution, which increases the receptive field of individual neurons while maintaining the resolution of the output feature map. Thirdly, L2 normalization is applied on the three-channel output of the network, so that the output on each pixel is enforced to be a unit vector, which represents the estimated normal. We calculate the cosine similarity between the estimated normal and ground-truth normal as follows:(1)$$L_{norm}=\frac{1}{k}\sum_{i\in K}\cos\left(N_{i}-{\tilde{N}}_{i}\right),$$
in which $L_{norm}$ denotes the normal estimation loss, *K* denotes the pixel set within the image, and *k* denotes the number of pixels in *K*. $N_{i}$ and ${\tilde{N}}_{i}$ denote the estimated normal and the ground-truth on ith pixel, respectively.

**RANSAC plane estimation**. We estimate the 3D plane where the transparent object is placed from the D-channel. We preprocess the original depth using the output of transparent segmentation. The depth of transparent region is discarded to eliminate the influence of depth error, and the depth of non-transparent region is retained for plane estimation. With the camera’s intrinsic parameter matrix, the depth is converted into 3D point cloud. We estimate the 3D plane where the transparent object is placed by a RANSAC (Random Sample Consensus) plane detection algorithm [54]. The detected plane is considered to be valid if the number of inlier points is more than a specific threshold. To ensure the robustness of plane estimation, we repeat the RANSAC plane fitting with rest points, and then the depth of the best fitted plane is selected to be fed into the second stage.

### 3.2. The Second Stage

The second stage takes the results of the first stage as input and then estimates the 6DoF object pose. Firstly, the color feature is extracted from cropped color patch. Secondly, the UV code, the surface normal, and the depth value on plane are concatenated to form a data structure named “extended point-cloud”. Thirdly, based on the extended point-cloud and the color feature, the 6DoF object pose is estimated by a DenseFusion-like network [2].

**Color Feature Extraction**. For each segmented object instance, we crop color patch $P_{color}$ from the RGB-channel. $P_{color}$ is resized to an uniform size of 80×80 for color feature extraction. The color feature extraction network is a CNN-based encoder–decoder architecture. It takes $P_{color}$ as input and outputs color feature with the same size as $P_{color}$. The dimensional of the output color feature is 32. We randomly sample 500 pixels within the segmented transparent region. For each sampled pixel *x* in the patch, the color feature on that pixel is $P_{color}\left(x\right)$. It is later fed into the DenseFusion-like network for 6DoF pose estimation.

**Extended point-cloud**. Ordinary methods [2,3] take classic point-cloud of depth as input for pose estimation. However, we do not use the point-cloud of original depth as input, as it has been dramatically distorted by depth error. We also do not reconstruct depth to obtain rectified point-cloud, because accurate depth reconstruction is time consuming. We rectify the distorted geometry by surface normal recovery. Since surface normal alone is insufficient for absolute pose estimation [12], an extended point-cloud representation is proposed to estimate 6DoF object pose in our research.

For each segmented transparent object instance, we crop patches from the UV encoding map, the estimated surface normal, and the depth of the estimated plane. The UV encoding map encodes the 2D (u,v) coordinates of each pixel on image. The estimated surface normal and depth of the estimated plane are the output of the first stage. The cropped patches $P_{UV}$, $P_{norm}$, $P_{plane}$, are resized to an uniform size of 80×80 for extended point-cloud construction.

For each sampled pixel *x* in the patch, the extended point-cloud is defined by concatenating $P_{UV}\left(x\right)$, $P_{norm}\left(x\right)$, and $P_{plane}\left(x\right)$. The roles of these three components are as follows: (1) $P_{UV}\left(x\right)$ indicates the 2D location of a pixel on image plane, and it is an important hint for 6DoF pose estimation [1,13]; (2) $P_{norm}\left(x\right)$ indicates the surface normal of object, and it is important for relative pose (3DoF rotation) estimation; and (3) $P_{plane}\left(x\right)$ provides the hint where the object is placed, and it helps conduct more accurate 3DoF translation estimation.

**6DoF Pose estimation.** Taking the extended point-cloud and the color feature as input, we adopt a DenseFusion-like network structure (similar to [2,3]) for 6DoF pose estimation. As illustrated in Figure 7, edge convolution [55] is applied on the extended point-cloud to extract geometry features. Then, the geometry and color features are densely concatenated for 6DoF pose estimation. We randomly sample *N* pixels for feature extraction. In this work, the value of *N* is 500, the same as that of Rotation Anchor [3] and DenseFusion [2]. The number of anchors is 60, the same as that of Rotation Anchor [3].

The network structure of ours is very similar to that of Rotation Anchor [3], and the difference is that Rotation Anchor takes 3D classic point-cloud as input but ours takes 6D extended point-cloud as input. Thus, the weight matrix of the first fully connected layer in the geometry feature branch is 6×64, and that of Rotation Anchor is 3×64. The pose estimation process is described as follows.

Instead of directly regressing the translation [56,57] or key-points [58,59] of object, predicting vectors that represent the direction from pixel toward object is more robust [3]. For each of the pixel, we predict the vector points to the center of the object and normalize it to a unit vector. This vector-field representation focuses on local features and is sensitive to occlusion and truncation. We use the unit vectors to generate coordinates of object center in a RANSAC-based voting scheme. With the coordinates of the center point, we can obtain the 3D translation of the object.

Due to pose ambiguity caused by symmetrical objects, directly regressing rotation is always a challenge for 6DoF pose estimation task. Early experimentation [11,60] shows clearly that using discrete-continuous regression scheme is effective to obtain accurate rotation. However, SSD-6D [60] requires prior knowledge of 3D models to manually select classified viewpoints, and Li et al. [11] did not enforce local estimation for every rotation classification. Similar to Tian et al. [3], rotation anchors are used to represent sub-part of the whole rotation space SO(3) which is divided equally. For each rotation anchor **${\hat{R}}_{j}$**, where the subscript *j* denotes the index of the anchor, we predict the rotation offset $▵R_{j}$ and the predicted rotation:(2)$$R_{j}=▵R_{j}{\hat{R}}_{j}.$$

Additionally, Confidence $C_{j}$ is predicted to represents the similarity between $R_{j}$ and the ground truth. The prediction of the anchor with the highest confidence is selected as the output. The anchor index is selected as $\hat{j}=\underset{j}{\arg\max}C_{j}$; the output 3DoF rotation $R=R_{\hat{j}}$, and confidence $C=C_{\hat{j}}$.

The loss $L_{pose}$ aims to constrain distance between the estimated pose and the ground truth pose. It is divided into three parts:(3)$$L_{pose}=\lambda_{3}L_{shape}+\lambda_{4}L_{reg}+\lambda_{5}L_{t},$$
where $L_{shape}$ is an extension of ShapeMatch-Loss [56] in the aspect of difference in object diameter. $L_{shape}$ normalize the loss with object diameter:(4)$$L_{shape}=\frac{\sum_{x_{1}\in M}\;\min_{x_{2}\in M}\;||\;Rx_{1}-\tilde{R}x_{2}\;{\left|\right|}_{2}}{m\times d\times C}+\log C,$$
where *m* denotes the number of points in 3D object model M, *d* denotes the diameter of object, $\tilde{R}$ denotes the ground truth rotation, and *C* denotes the confidence. The loss $L_{reg}$ constrains the regularization range of $▵R_{j}$,
(5)$$L_{reg}=\sum_{j}\max\left(0,\max_{j\neq k}<q_{j},\hat{q_{k}}>-<q_{j},\hat{q_{j}}>\right)\;,$$
where $q_{j}$ and $\hat{q_{j}}$ are quaternion representations of $R_{j}$ and ${\hat{R}}_{j}$. The closer the estimated rotation is to the ground truth rotation, the larger the dot product of the two quaternion matrices becomes. The loss $L_{t}$ constrains the translation of object, and it is calculated as the smooth L1 norm distance between the RANSAC-based voting results and the ground-truth translation.

### 3.3. Dataset

We evaluated our approach on ClearGrasp dataset [12] which contains five common transparent objects with symmetric properties. Occlusions and truncation exist in the images captured, which make this dataset challenging. As far as we know, it is the only publicly available RGB-D dataset applicable for transparent object 6DoF pose estimation. ClearGrasp dataset contains synthetic images and real images. For the quantitative experiments (Section 4.4, Section 4.5 and Section 4.6), we used the synthetic data in which the ground-truth pose is available. For the qualitative experiments (Section 4.6), we used the real images and synthetic images. From the synthetic data, we randomly picked 6000 images for training and 1500 images for testing. Each image contains multiple annotated objects. For the training set, there are 14,716 instances, and, for the testing set, there are 4765 instances. All the compared methods were trained and tested using the same training and testing split. DenseFusion [2] and Rotation Anchor [3] were trained and tested using the depth generated by the compared depth completion methods. For initialization, we used the pretrained model publicly available, and then fine-tuned the network using the training data. In the experiments, we adopted the standard depth completion pipeline of ClearGrasp [12].

## 4. Experiments

### 4.1. Experimental Settings

All experiments were performed on a computer with an Intel Xeon Gold 6128 CPU and a NVIDIA TITAN V GPU. The proposed approach was implemented with PyTorch [61]. The network was trained using an Adam optimizer [62] with an initial learning rate of 0.001. The batch size was set to 4, and the images were resized to a resolution of 640 × 480. We adopted intermediate supervision to solve vanishing gradients problem which calculates losses at different stages of the network. The total loss is $L=\lambda_{1}L_{norm}+\lambda_{2}L_{pose}$, where $L_{norm}$ is used to constrain the surface normal estimation and $L_{pose}$ is used to constrain the final 6DoF pose estimation. We set the parameter $\lambda_{1}$ to decrease from 1 to 0, and then freeze the weights of normal estimation sub-network, while we set the parameter $\lambda_{2}$ to increase from 0 to 1. $L_{pose}=\lambda_{3}L_{shape}+\lambda_{4}L_{reg}+\lambda_{5}L_{t}$. Following Tian et al. [3], $\lambda_{3}$, $\lambda_{4}$, and $\lambda_{5}$ were set as 1, 2, and 5, respectively. The training stopped when the number of epochs reached 80.

### 4.2. Evaluation Metric

Typically, ADD metric [63] is used to evaluate pose error for asymmetric objects, and ADD-S metric [56] is used for both symmetric and asymmetric objects. Given the ground truth pose **(R,T)** and the estimated pose **$\left(\tilde{R},\tilde{T}\right)$**, ADD metric is defined as the average distance between the corresponding points after applying the transformations to the 3D model:(6)$$\mathrm{ADD}=\frac{1}{m}\sum_{x\in M}\;||\;\left(Rx+T\right)-\left(\tilde{R}x+\tilde{T}\right)\;\left|\right|,$$
where *x* denotes a point within the 3D object model M and *m* denotes the number of points within the model.

For symmetric object which has multiple visually correct poses, ADD will cause many misjudgments. As only one of the correct poses is labeled as ground-truth, the estimation is considered as correct only if it matches with the ground-truth pose. However, other visually correct poses will be judged as false estimations. To resolve this problem, the overall evaluation of both symmetric and asymmetric objects are taken into account by ADD-S metric. ADD-S is defined as the average distance to the closest model point:(7)$$\mathrm{ADD}-S=\frac{1}{m}\sum_{x_{1}\in M}\;\min_{x_{2}\in M}\;||\;\left(Rx_{1}+T\right)-\left(\tilde{R}x_{2}+\tilde{T}\right)\;\left|\right|.$$

In our experiments, we use ADD-S metric for evaluation, because transparent object are commonly symmetric. The estimated pose is considered as correct if ADD-S is less than a given threshold. We set the threshold to 10% of the object diameter, the same as that of many previous works [2,3].

### 4.3. Accuracy

We compared our approach with state-of-the-art baselines. Generally, to eliminate the depth error of transparent object for accurate 6DoF pose estimation, it is straight-forward to reconstruct the depth of transparent object first, and then predict the 6DoF pose using ordinary RGB-D pose estimation methods. For the depth reconstruction, we compared two options: (a1) *FCRN* denotes Fully Convolutional Residual Networks for depth reconstruction [64]. It directly estimates depth using 2D CNN-based network from RGB-channel. (a2) *CG* denotes ClearGrasp [12], an accurate depth reconstruction algorithm. It estimates the surface normal from the RGB-channel, and then reconstructs the depth by an optimization scheme. For the RGB-D pose estimation methods, we considered two state-of-the-art methods: (b1) *DF* denotes DenseFusion [2]. It densely fuses the color and geometry features for accurate pose estimation. (b2) *RA* denotes Rotation Anchor [3]. It is one of the most stable and accurate pose estimation methods for both symmetric and asymmetric objects. By combining the options mentioned above, we had four state-of-the-art baselines: (1) *FCRN* [64] + *RA* [3]; (2) *FCRN* [64] + *DF* [2]; (3) *CG* [12] + *DF* [2]; and (4) *CG* [12] + *RA* [3]. In the following, *Ours* denotes the proposed approach.

The accuracy–threshold curves of the compared methods are shown in Figure 8. The x-axis denotes the varying threshold in terms of ADD-S, and the y-axis denotes the accuracy according to the threshold. The AUC is the area under the accuracy–threshold curve within the range from 0 to 0.1 m. The accuracy of the compared methods are shown in Table 1, in which the threshold is set to 10% of the object diameter. We observed that *FCRN*-based methods can efficiently reconstruct depth, but the result is not accurate, as the depth is inferred from RGB-channel only. *CG*-based methods can accurately reconstruct depth, because both RGB-channel and D-channel are considered in the depth reconstruction process. *DF*-based methods are stable, even when the estimated depth is not accurate; *FCRN* + *DF* still yields sensible result. *RA*-based methods are accurate; *CG* + *RA* outperforms *CG* + *DF* when the reconstructed depth is good, but the performance of *FCRN* + *RA* degrades when the depth reconstruction is inaccurate. *Ours* does not rely on depth reconstruction as it directly estimate object pose from estimated normal map, and experimental results show that it is the most accurate among the compared methods.

### 4.4. Efficiency

The time efficiency of the compared methods are shown in Table 2. We observed that the *FCRN*-based methods are efficient but not accurate, because the depth is reconstructed from RGB-channel only. The *CG*-based methods take much longer than *FCRN*-based methods, since accurate depth is constructed by *CG* through a time consuming global optimization scheme. The *RA*-based methods are slightly less efficient than the *DF*-based ones, as the network structure of *RA* is more complex than *DF*. Among the compared methods, Ours is the most efficient, because the depth is not reconstructed and the estimation is directly conducted based on extended point-cloud. The time efficiency of *Ours* is 0.069 s per instance, and 0.223 s per image (a single image may contain multiple object instances).

### 4.5. Ablation Study

The extended point-cloud contains three components: the UV code, the normal, and the plane. To study the importance of these components, an ablation study was conducted by evaluating three variations of the extended point-cloud: (1) *w/o UV code*, which denotes the extended point-cloud without the UV code component; (2) *w/o normal*, which denotes the extended point-cloud without the normal component; and (3) *w/o plane*, which denotes the extended point-cloud without the plane component.

The 6DoF pose estimation accuracy of the three variations are shown in Table 3, and the accuracy–threshold curves of the three variations are shown in Figure 9. The observations are as follows: (1) UV encoding is crucial for extended point-cloud. When the UV code is absent, the accuracy of *Ours* drops from 85.4% to 33.8%. The pose estimator cannot properly work without UV code. (2) Normal estimation is very important for accurate 6DoF transparent object pose estimation. When the normal is absent, the accuracy of *Ours* drops from 85.4% to 68.8%. (3) Plane estimation is helpful to improve the accuracy. When the plane is absent, the accuracy of *Ours* drops from 85.4% to 71.5%.

It is also observed that the normal component is closely related to the 3DoF rotation estimation accuracy. To measure the 3DoF rotation accuracy, we replace the estimated translation with ground-truth, and then measure the ADD-S accuracy. The 3DoF rotation accuracy of *Ours* is 95.2%. It drops to 83.9% when normal is absent, while it slightly drops to 92.0% when plane is absent. It shows that the normal component is more closely related to the 3DoF rotation estimation.

### 4.6. Qualitative Evaluation

The 6DoF pose estimation results of the compared methods are visualized in Figure 10. Each row corresponds to a scene, and each column corresponds to a method. The first column is the results of *CG + DF*, the second column is the results of *CG + RA*, and the third column is the results of *Ours*. The object pose is shown as tight oriented bounding box of object’s model. Different objects are visualized by boxes with different colors. The results show that *Ours* accurately estimates transparent object pose in simple scenes (Rows 1 and 6), cluttered scenes (Rows 2 and 4), with occlusions (Rows 4 and 5), and small scale size (Row 6). We also predict truncated objects (partially visible) more accurately than other methods, e.g., the object in the Row 7. Our approach is trained using synthetic images, but the trained model can be robustly applied to real images which are not seen in the training data (please refer to the first four rows of Figure 10).

When several transparent objects occlude each other, the pose estimation results may be unstable in some extent. The failure cases are shown in Figure 11. In Figure 11a, when one transparent object is occluded by another transparent object, the correctness of pose estimation may be affected. In Figure 11b, when multiple small transparent objects overlap each other, several small objects may be falsely estimated as one. As the light paths within mutually occluding transparent objects are complicated, it is very hard to recover the geometry of transparent object behind the transparent object. We will consider to research this complex issue in the future.

## 5. Conclusions

In this paper, we present a two-stage approach for 6DoF pose estimation of transparent object from a single RGB-D image. The missing or distorted geometric information caused by depth error is recovered in the first stage, and then the 6DoF pose of transparent object is estimated in the second stage. Specially, an extended point-cloud representation is proposed to estimate object pose accurately and efficiently without explicit depth reconstruction.

In this study, the device we use is the Intel RealSense D435 camera, the same as that of ClearGrasp. The proposed approach is trained and evaluated on the ClearGrasp dataset. It is an interesting topic to discuss whether the proposed approach can be applied to different RGB-D devices. We believe that the proposed approach will work when the following assumptions are satisfied: (1) the RGB-channel and the D-channel are pixel-wisely aligned; (2) the transparent object stands on an opaque plane; and (3) the depth of the opaque plane can be reliably captured by the depth sensor.

If the first assumption is satisfied, no matter what type of depth sensor is used, the noisy depth of the transparent object can be effectively removed, since the segmentation is predicted using only RGB-channel. If the second and third assumptions are satisfied, the plane can be effectively estimated for extended point-cloud extraction. Fortunately, most of the consumer level RGB-D devices satisfy the first and third assumptions. The second assumption holds in many common application scenarios in our daily life. If the above three assumptions are not met, the proposed approach will degrade to the baseline “w/o plane” addressed in Section 4.6. The pose estimation accuracy of “w/o plane” is 71.5. “w/o plane” is less accurate than “Ours”, but it still performs quite well comparing to other state-of-the-art baselines.

Furthermore, we have some notes about the compared baselines: *FCRN* [64] is a general-purpose depth estimator which takes only RGB-channel as input. *CG* [12] is an impressive transparent depth recovering method which generates high accurate depth map for robotic grasping, and it can be generalized to unseen objects. *DF* [2] and *RA* [3] are among the state-of-the-art ordinary object pose estimators. Surprisingly, the experimental results show that the combinations of these advanced methods do not work very well for 6DoF transparent object pose estimation. The observation suggests that accurate transparent depth estimation with accurate ordinary object pose estimation does not necessarily result in accurate transparent object pose estimation. In this paper, we focus on the specific challenge in the 6DoF transparent object pose estimation problem, and the proposed approach achieves significantly more efficient and accurate performance than the state-of-the-art ones by utilizing the novel extended point-cloud representation. Different from the grasping task [12] which *CG* aims at, we focus on the 6DoF pose estimation task, which normally assumes that 3D models of objects are known [2,3]; therefore, unlike *CG*, the proposed approach cannot be applied to unseen objects.

The limitation of the work is that the pose estimation accuracy may degrade when multiple transparent objects occlude each other (i.e., transparent object occludes transparent object), which is a very challenging scenario. In the future, we will apply this technology in robotic teaching and manipulation applications and study the interaction between human hand and the transparent objects.

## Figures and Tables

**Figure 1 sensors-20-06790-f001:**
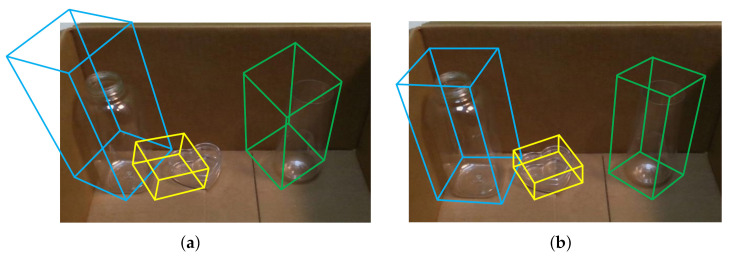
Pose estimation results of transparent objects. (**a**) DenseFusion [2], (**b**) Our proposed approach. The object pose is expressed as tight oriented bounding box of the object model. DenseFusion [2] is one of the state-of-the-art methods for 6DoF object pose estimation from a single RGB-D image.

**Figure 2 sensors-20-06790-f002:**
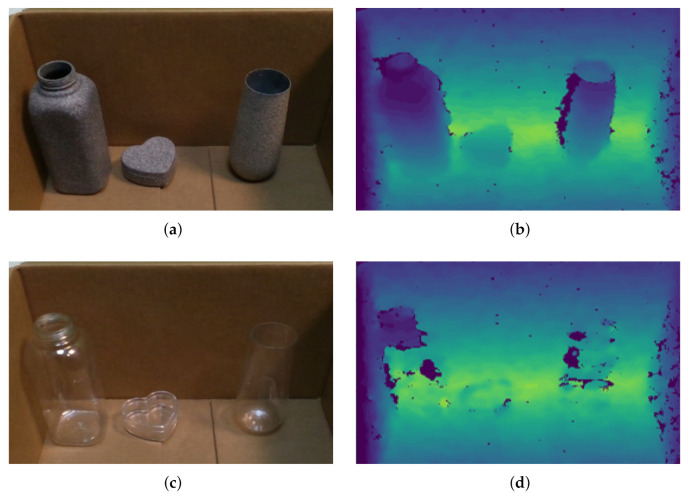
Depth errors of transparent objects: (**a**) RGB-channel of non-transparent objects; (**b**) D-channel of non-transparent objects; (**c**) RGB-channel of transparent objects; and (**d**) D-channel of transparent objects. For the transparent objects, there exist two types of depth error in the D-channel. The depth error dramatically distorts the 3D geometric information of transparent objects observed.

**Figure 3 sensors-20-06790-f003:**
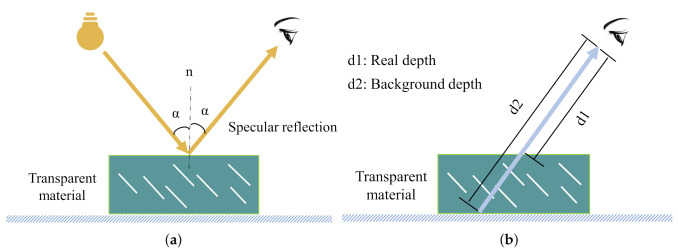
Cause of depth error of transparent material: (**a**) depth error of Type I is caused by specular reflection; and (**b**) depth error of Type II is caused by light passing through transparent material.

**Figure 4 sensors-20-06790-f004:**
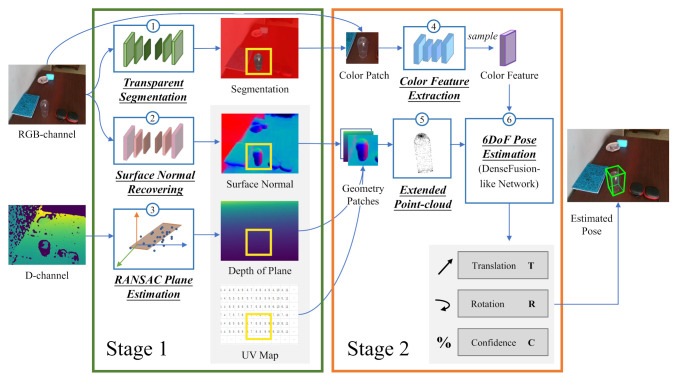
Framework of our proposed approach. In the RANSAC Plane Estimation step, the noisy depth in the segmented transparent region is removed before plane estimation. After the Color Feature Extraction step, 500 pixels are randomly sampled in the segmented transparent region.

**Figure 5 sensors-20-06790-f005:**
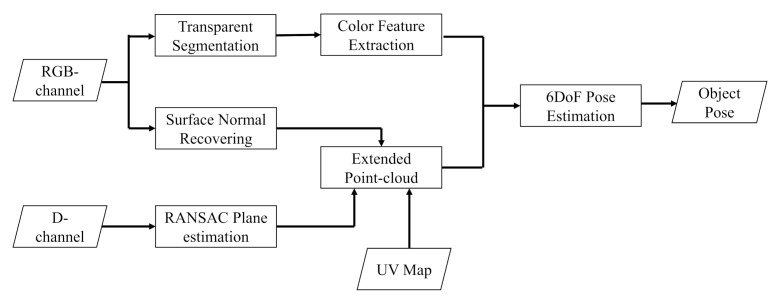
Block-diagram of our proposed approach.

**Figure 6 sensors-20-06790-f006:**
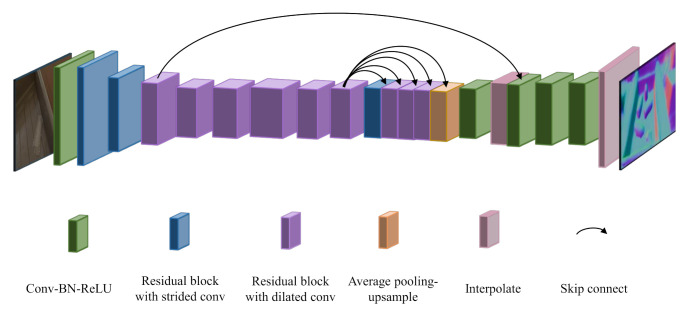
Surface normal estimation network structure.

**Figure 7 sensors-20-06790-f007:**
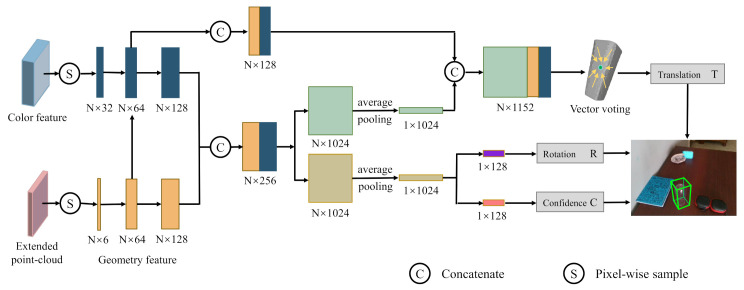
Pipeline of the 6DoF pose estimation network.

**Figure 8 sensors-20-06790-f008:**
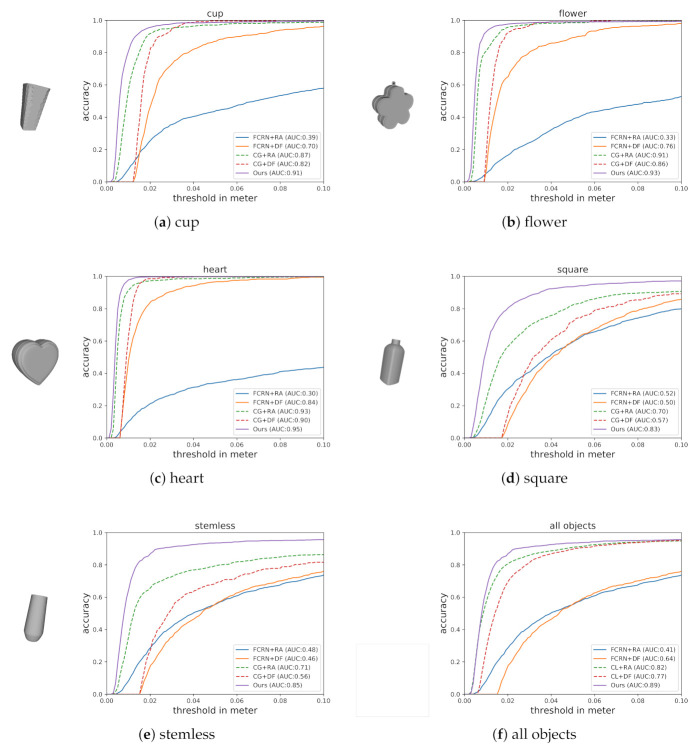
3D model of each object and the accuracy–threshold curves of the objects.

**Figure 9 sensors-20-06790-f009:**
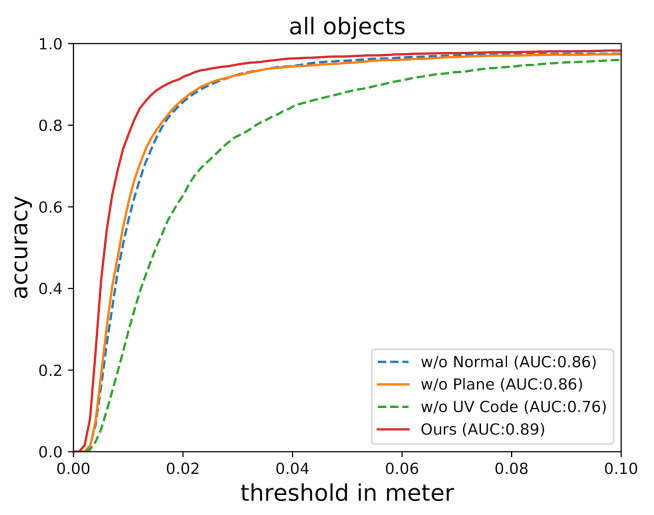
Ablation study. The accuracy–threshold curve of the three variations.

**Figure 10 sensors-20-06790-f010:**
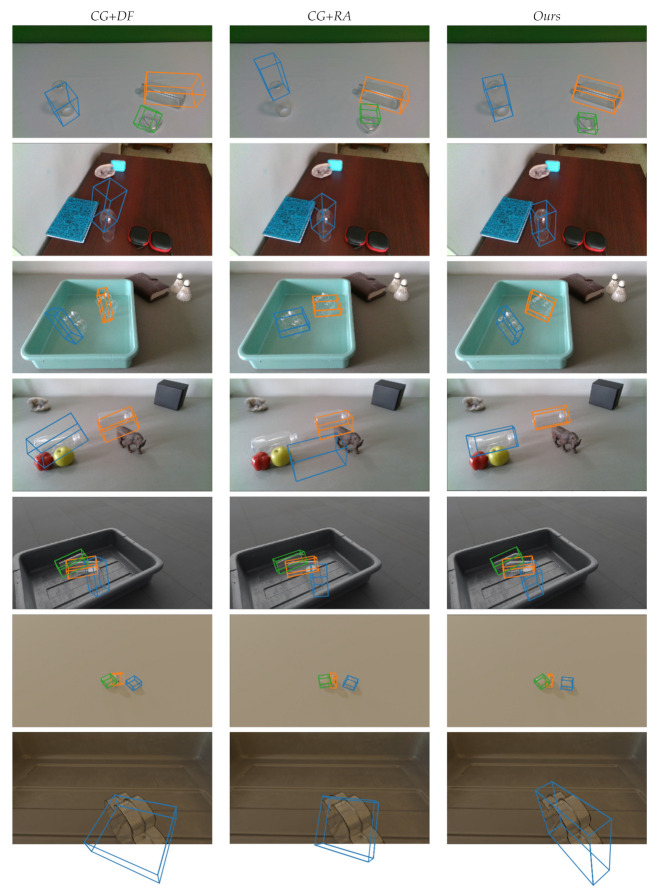
The transparent object pose estimation results of *CG+DF*, *CG+RA*, and *Ours*. The object poses are visualized as tight oriented bounding boxes. The first four rows are real images, and the last three rows are synthetic images.

**Figure 11 sensors-20-06790-f011:**
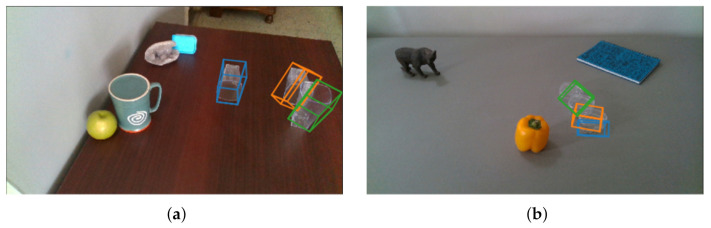
Failure cases: (**a**) a transparent object occludes anothor, (**b**) several small transparent objects occlude each other.

**Table 1 sensors-20-06790-t001:** Pose estimation accuracy of the compared methods. The threshold of ADD-S is set to 10% of the object diameter. Each row corresponds to a type of object. The last row is the evaluation results of all objects. The bold font indicates the best scores.

Object	*FCRN* + *RA*	*FCRN* + *DF*	*CG* + *DF*	*CG* + *RA*	*Ours*
cup	12.2	40.3	76.5	71.2	**88.6**
flower	13.5	53.8	76.3	80.3	**92.2**
heart	7.7	35.2	28.5	73.3	**88.7**
square	25.5	35.6	71.8	54.4	**77.0**
stemless	23.2	32.4	69.3	62.7	**80.1**
all	16.4	39.1	64.0	68.4	**85.4**

**Table 2 sensors-20-06790-t002:** Time efficiency of the compared methods.

	*FCRN* + *RA*	*FCRN* + *DF*	*CG* + *DF*	*CG* + *RA*	*Ours*
per instance	0.108 s	0.074 s	0.819 s	0.855 s	0.069 s
per image	0.345 s	0.234 s	2.606 s	2.715 s	0.223 s

**Table 3 sensors-20-06790-t003:** Ablation study. Pose estimation accuracy of the three variations. The threshold of ADD-S is set to 10% of the object diameter.

Object	*w/o UV Code*	*w/o Normal*	*w/o Plane*	*Ours*
cup	35.6	71.6	72.5	88.6
flower	44.0	79.7	81.0	92.2
heart	36.2	69.4	73.4	88.7
square	23.8	58.1	62.3	77.0
stemless	29.2	65.3	68.3	80.1
all	33.8	68.8	71.5	85.4

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
