# Peer review of "6DoF Pose Estimation of Transparent Object from a Single RGB-D Image"

_sensors, 2020, doi:10.3390/s20236790_

Round 1

Reviewer 1 Report

The authors present a method for 6DoF pose estimation of objects using RGB+Depth images. The topic is of scientific interest and the authors report improved results in their state-of-the-art comparison. However, the paper leaves unanswered questions about the methodology and experiments which need to be addressed before it is considered for publication.

- Scope. There are different existing depth sensing technologies. A different behaviour can be expected when sensing transparent objects with different technologies. The authors do not seem to take this into account in their approach. They mention using RGB-D data from the ClearGrasp paper so we can assume that their method works for the specific camera used in that dataset. Some further insights of the effects of the device would be welcome to increase the scope width of the article. The authors mention that they ignore approaches using other technologies in the related work section but they do not even describe the device used for their experiments. There could be a large difference in result even between a kinect I and II RGB-D device.   

Novelty: The proposed pipeline is mostly a combination of different existing components. The authors should be more clear about the specific changes they introduce to the components. More in detail:

  1. The claim that this method is the first that deals with the problem of transparent 6DoF object pose is repeated often and is a bit too strong here in my opinion considering that the problem can be addressed by a sequential combination of existing methods as shown in the comparison section (ClearGrasp+DenseFusion etc).
  2. The term "DenseFusion-like" network is used often by the authors. For clarity it would be better to mention using this network with the specific modifications they have done, if any.
  3. It is not clear, which pipeline components are novel. Very importantly citations are missing when describing the first and second stage of the approach. Surface Normal recovering and RANSAC plane estimation are well known methods but there is no citation on the corresponding sections. This is misleading.

Method and Experiments: Parts of the method need to be explained better and further details of the experiments are needed to ensure fairness.

  1. Transparent segmentation. This is an important part of the pipeline but the authors only mention using Mask R-CNN. More information is needed about how it is trained, how many classes are detected, how the performance of this impacts the rest of the pipeline. If the method is used as-is from previous works it should be cited.
  2. Experiments. First of all, the authors make an arbitrary choice of a training-testing set from the ClearGrasp dataset. This dataset contains synthetic and real images and the method is trained on the synthetic images. It should be clarified which images are used here. More importantly, it should be clarified whether all the methods the authors compared to were retrained on the same data or used as pretrained network models. This raises an issue of fairness in the comparison. The same holds for the time efficiency comparison. Which hardware was used for the runtime comparison should be mentioned.

Related Work comparison. The authors compare to ClearGrasp as the most related work to theirs. They should mention that ClearGrasp is trained on synthetic data which is more challenging, it reconstructs dense depth of the scene which is useful for grasping tasks and it is able to generalize to unseen objects. How does this work compare in these domains?

Writing. The paper is in general well written, a spellcheck for typos should be done. A few sections are in my opinion less well written and some effort should be spent to improve them. I would suggest this for Section 2 lines 74 to 86 and the experiments sections.

Reviewer 2 Report

This paper proposes a novel idea for 6DoF Pose Estimation of Transparent Objects.

However, these has some questions and comments for adjustments. According to the organization of manuscript the comments were as follows:

Major comments:

  1. The authors build a segmentation network to extract the transparent objects on the input RGB images in the Stage 1, as shown in Figure 4. However, you just get a color patch with many pixels of backgournd, so can an object detection network accomplish this task instead of a more complex segmentation network?
  2. As we know, the refraction of light is very common for transparent objects. I wonder whether it has some influence in this task.
  3. Based on your introduction of the method proposed in this paper, the Stage 1 must estimates a surface normal map which will be used in Stage 2. I have three questions about this point: The first is, how does the model learn the information of surface normal? Does it depend on the dataset? Second, will it fail to work in the environment without any opaque surface? The last is, even if there is a plane, but the plane is transparent, can it still work?
  4. This paper only estimates the pose of the object and ignores the category label of the predicted object. In other words, if there is an instance of an object in the RGB-D image which has never been seen before, how to estimate its 6-DoF pose?
  5. The pose estimation accuracy may degrade when multiple transparent objects occlude each other (i.e. transparent object339 occludes transparent object), which is a very challenging scenario. How to solve this problem?
  6. The author can consider the pose estimation of all instance objects rather than just transparent objects.

Minor comments

  1. In this paper, sometimes the past tense is used,but sometimes the present tense is used. Please adjust this.
  2. In Figure 6, I am confused about the network structure. For example, I don’t understand what the N is and what the meaning of 32,64,128 is. Are they the feature maps from CNN or Fully connected layers? In addition, can you provide a table to show the parameters used in this network?

Round 2

Reviewer 1 Report

The authors took the review comments into account and provided detailed explanations to the raised points.

My final impression of the paper is that it is written well, is methodologically sound and provides enough experimental results with good performance compared to the state-of-the-art. The proposed method is well engineered. The paper is of limited scientific novelty. As confirmed by the authors answers, it consists mainly of a combination of existing components.

Since the method might be of interest as a full pipeline for transparent object pose estimation, my final rating is towards (weak) accept.

Author Response

Thanks for all your helpful suggestion!

Have a nice day!

Reviewer 2 Report

  1. Please prepare a brief block-diagram regarding the proposed mathod in the introduction, which can make the reader understand the article more clearly. Then explain about sub-blocks briefly.
  2. There is still a small problem I care about whether the background pixels of color patch in Figure 4 have an impact. In other words, I have known that you use segmentation network to obtain the mask of the object, and I think you will use the mask to eliminate the background pixels,right?It is might be better to explain it clearly in Figure 4.
  3. I suggest to represent only results of the experiment. Dataset can be inserted in previous section. 
  4. How can you validate the robustness of proposed method?
  5. Conclusion is a summary of the paper.  Moreover, authors have to extend the discussion about limitations of this study concerning methods applied, measures, data available and application in different contexts. However, there are no discussion about this.
